# Designing of Hollow Core Grapefruit Fiber Using Cyclo Olefin Polymer for the Detection of Fuel Adulteration in Terahertz Region

**DOI:** 10.3390/polym15010151

**Published:** 2022-12-29

**Authors:** Sakawat Hossain, Md. Aslam Mollah, Md. Kamal Hosain, Md. Shofiqul Islam, Abdulhameed Fouad Alkhateeb

**Affiliations:** 1Department of Electronics & Telecommunication Engineering, Rajshahi University of Engineering & Technology, Rajshahi 6204, Bangladesh; 2Department of Electrical and Computer Engineering, Faculty of Engineering, King Abdulaziz University, P.O. Box 80204, Jeddah 21589, Saudi Arabia

**Keywords:** photonic crystal fiber, finite element method, zeonex, terahertz, sensor

## Abstract

A grapefruit-shape hollow-core liquid infiltrated photonic crystal fiber (LI-PCF) is proposed and evaluated to identify the percentage of kerosene in adulterated petrol. The proposed hollow-fiber sensor is designed with Cyclo Olefin Polymer (Zeonex) and likely to be filled with different samples of petrol which is adulated by the kerosene up to 100%. Considering the electromagnetic radiation in THz band, the sensing properties are thoroughly investigated by adopting finite element method (FEM) based COMSOL Multiphysics software. However, the proposed sensor offers a very high relative sensitivity (RS) of 97.27% and confinement loss (CL) less than 10^−10^ dB/m, and total loss under 0.07 dB/cm, at 2 THz operating frequency. Besides that, the sensor also possesses a low effective material loss (EML), high numerical aperture (NA), and large Marcuse spot size (MSS). The sensor structure is fabrication feasible through existing fabrication methodologies consequently making this petrol adulteration sensor a propitious aspirant for real-life applications of petrol adulteration measurements in commercial and industrial sensing.

## 1. Introduction

Over the past few years, researchers have shown a significant amount of interest in terahertz (THz) based waveguides and functional devices for their promising and potential applications. The THz frequency spectrum, which ranges from 0.1 THz to 10 THz or 0.03 mm to 30 mm, and is located between the microwave and infrared bands. The widespread applications of THz technology have evolved from conventional fields of astronomy to new fields such as terahertz communication [1], time-domain spectroscopy [2], sensing [3], military works [4], medicine [5], drug testing [6], etc. PCFs operating at the terahertz (THz) frequency regime have attained notable advancement in the fields of biomedical applications in recent years. This is because THz radiation has no detrimental effects on human health over X-rays as well as a non-ionizing characteristic on organic cells [7], making THz PCF one of the resourceful approaches for applications in real-time detection of cancer vs. non-cancer cells [7], colon cancer [8,9], diagnosis of skin cancer [10], tumor detection [11], etc.

The properties of PCFs, such as birefringence, dispersion, effective material loss (EML), single-mode propagation, transmission spectra, effective area (EA), and so on, could be modified in ways by properly tuning the shape and diameter of air holes used in both the core and cladding areas, as well as the pitch (distance from center to center between two consecutive air holes) which are impossible to achieve with traditional solid core fiber [12]. In addition, PCF shows robustness under harsh conditions including noise, high voltage, high temperature, powerful chemical, and electromagnetic fields [13]. Due to these advantageous properties over traditional fiber, PCFs are more appealing to researchers for a wide variety of applications including terahertz communication [1,14] image rebuilding [15], filters [16], parametric amplification variable optical switches [17,18] and so forth. Furthermore, hollow core PCF facilitates the infiltration of analytes (chemicals, gases, etc.) into the air-core region, allowing for strong analyte-light interaction and paving the way for new sensing applications [19]. Due to these tempting capabilities, hollow core PCFs have been utilized to develop numerous sensors that includes gas sensing [20], magnetic field measurement [21], refractive index sensor [22], chemical sensing [23], temperature sensor [24], curvature sensor [25], blood component detection [26], micro-displacement sensor [27], etc.

As a growing field, PCF-based chemical analytes sensing has drawn significant interest among researchers worldwide. For example, in 2016, Arif et al. [28] demonstrated a hexagonal photonic crystal fiber (H-PCF) for sensing of chemical analytes and achieved relative sensitivity (RS) of 50%, 55.83%, and 59.07%, and confinement loss (CL) of 4.25 × 10^−10^ dB/m, 8.72 × 10^−10^ dB/m, and 2.56 × 10^−10^ dB/m for water, ethanol, and benzene, respectively. Later in 2018, Sultana et al. [29] introduced a PCF-based THz sensor for only alcohol detection. Their configured chemical sensor exhibits high RS of 68.87% and CL as low as 7.79 × 10^−12^ cm^−1^. Following the year, Paul [30] proposed a micro-structured quasi PCF for better detection of chemical analytes. The numerical analysis reveals that the quasi PCF sensor conveys improved RS responses of 78.8%, 77.8%, and 69.7% for ethanol, benzene and water, respectively. In addition, in 2018, Islam et al. [31] demonstrated asymmetrical hollow-core PCF for water, ethanol, and benzene detection and attained a comparatively higher RS of 96.69%, 96.97% and 97.2%, respectively, at 1.4 THz operating frequency. Hasan et al. [32] and Rahaman et al. [33] also proposed asymmetrical hollow-core PCF and achieved RS of 98% and 94.03%, respectively. Recently, negative curvature hollow core fiber has also been used for chemical sensing in THz region [34,35].

At the moment, fiber optic-based sensors are often used to detect fuel adulteration. Adulterated fuels are those that have been polluted or have had their content reduced as a result of the combination of low-quality chemicals. Since kerosene is less expensive than petrol, it is frequently mixed in with petrol to save money. The bad practice of fuel adulteration is still present as it is a profit-intended process. Detection of fuel adulteration is indispensable because machines running on adulterated fuel are mainly accountable for huge emissions of carbon dioxide (CO_2_), oxides of nitrogen, carcinogenic hydrocarbons, and other hazardous gases in the atmosphere. As a result, these harmful byproducts eventually dissolve into water, air, and soil consequently causing environmental pollution [36] and several diseases on human health [37]. Moreover, automobiles that run on adulterated fuel typically suffer from component breakdown or gradual degradation of engine parts more often than others [38]. Therefore, adulteration of fuels has now been an important issue. Numerous research works have been carried out earlier in regards to the detection of fuel adulteration. There are several methods by which adulteration levels of fuels by kerosene can be evaluated such as titration procedure [39], ultrasonic technique [40], etc. However, these procedures suffer from a series of limitations namely bulky and cumbersome equipment, often very tedious and time-consuming analytical methods, low level of precision and sensitivity, etc. Contrarily, surface plasmon resonance (SPR) gas earned a unique placement among sensing techniques due to its high accuracy, sensitivity, and unwavering approach. Utilizing this SPR technique, in 2019, Kawsar et al. [41] proposed a gold-coated PCF biosensor for oil adulteration identification. They have considered different mixtures of adulterated oil (10–50% concentration of kerosene in petrol) and based on the finite difference method (FDM) their simulation result shows a high RS of 93.5% at the optimum wavelength. Later in 2020, Habib et al. [42] proposed a rectangular shaped hollow-core PCF modeled and numerically evaluated in the THz band. In their study, they have used the rectangular hollow channel not only to fill different mixtures of adulterated fuel (0–100% concentration of kerosene in petrol) but also for driving the electromagnetic signal of the THz band. 

In this work, a noble hollow-core grapefruit-shape PCF is proposed for the better detection of adulterated petrol by kerosene in the THz frequency band. Here, two different core configurations (hexagonal and circular) with the same air-hole cladding are considered for sensor structure where the core is proposed to be filled with adulterated fuel samples of petrol. The main intention of this work is to propose a fabrication feasible sensor with improved sensing performance. There are considerable amounts of polymer materials e.g., silica, Topas, Teflon (Tetrafluroethylyne), PMMA (Polymethayl-methacrylate), HDPE (High Density Poly Ethylene), and Cyclo Olefin Polymer (Zeonex), etc. are available for a sensor to be designed. However, for THz waveguide, the background material of the fiber needs to be selected carefully. The most dominant type of loss of the fiber known as EML mainly depends on the bulk material of the fiber. We use Zeonex as the background material for its several advantages such as a uniform refractive index in THz frequency, excellent optical stability, and material dispersion, etc. over other polymers [43]. By transmitting the electromagnetic signal of the THz band through the core filled with the specimen of different concentration of adulterated petrol, the sensing characteristics of the proposed sensor is properly investigated. The outcomes of the numerical calculations reveal that the proposed sensor is capable of achieving a very high RS of 91.76% to 97.27% for different concentrations of adulterated petrol. The other parameters related to sensing characteristics are also investigated and offer better performance compared to other prior sensors. Moreover, this sensor can be a promising candidate in the detection of petrol adulteration in THz technology.

## 2. Sensor Geometry and Methodology

Figure 1a,b depicts the 2D view of the proposed hollow-core LI-PCF with circular and hexagonal core, respectively, and the corresponding 3D view is shown in Figure 1c,d, respectively. We have considered a grapefruit hollow-core PCF with two different cores such as circular and hexagonal configurations. The diameter of the both considered circular and hexagonal core and air hole are denoted as D_C_ and D, respectively. The adjacent gap or core struts between two consecutive air holes is represented as t and the gap between core and air holes is denoted as z. Figure 1e,f depicts the mesh representation of circular and hexagonal core, respectively. For the circular core, the overall mesh area is 2.618 × 10^−6^ m^2^ which is mainly composed of 35,942 triangular elements, with 90 and 1889 vertex and edge elements, respectively, while the overall mesh area is 2.618 × 10^−6^ m^2^ which is mainly composed of 51,946 triangular elements, with 116 and 2474 vertex and edge elements, respectively, for the hexagonal core. The overall fiber diameter of the proposed grapefruit shape hollow-core PCF for both circular and hexagonal core configurations is 1.66 × 10^−3^ m. A perfectly matched layer (PML), which absorbs the electromagnetic waves propagating outwards and inwards to the fiber’s outer surface [44] was imposed at the outermost circumference of the proposed fiber with a depth of 10% of the overall fiber diameter.

Figure 2 depicts an operational approach for the pragmatic execution of the proposed sensor. In order to have interactions with the sample under test (SUT), the light either broadband or supercontinuum depending on the type of source must be passed through the waveguide emitted from an optical tunable source (OTS). The emitted light should be linearly polarized before entering into the single mode fiber (SMF). This is carried out by passing the emitted light from OTS via polarizer and polarizer controller (PC). A mass flow controller (MFC) can be used to manage the sample flow rate into the C-type fiber. The C-type fiber helps to fill the core of the PCF with fuel. Then in the core region of the proposed sensor, there happens to be a strong light interaction with different concentrations of adulterated petrol. For the extraction of fuel from the proposed sensor, another C-type fiber should be connected. Before the light passes through an optical spectrum analyzer (OSA), another SMF must be used to collect the light after interacting with fuel. Finally, the waveforms can be sensed by a computer using OSA.

The proposed grapefruit-shaped hollow-core PCF must be feasible for fabrication in order to have practical applications such as commercial and industrial sensing. There are many commercially available fiber fabrication methodologies such as sol-gel [45], capillary stacking [46], drilling [47], stack and draw [48,49], extrusion [50], 3D-printing [51], etc. that have been used for an extended period for fiber production. As shown in Figure 1, our proposed grapefruit hollow-core PCF consists of six grapefruit-shaped air holes in the cladding with two considered hollow-core such as circular and hexagonal core structures. Since any types of complex e. g symmetrical and asymmetrical structures of PCF are likely used to fabricate employing the extrusion and 3D- printing techniques [50,51], the grapefruit-shaped air-holes of the proposed sensor can be fabricated using extrusion methodology. Fabrication of the grapefruit-shaped air-hole structure is possible as described in [52]. The considered two different cores configuration can also be fabricated through the existing fabrication process. For circular core consideration, it can be easily fabricated using stack and draw or so-gel techniques and the rest hexagonal core can be fabricated using extrusion methodology. The hexagonal core can also be fabricated by adopting rotational casting and rod in tube methods as described in [53]. Therefore, the proposed grapefruit hollow-core LI-PCF is practically feasible using the existing fabrication methodologies. In recent days, PCFs are commonly filled with liquid with the help of femtosecond laser [54], focused ion beam [55], and PBG fiber [56], and are consistently used in the laboratory environment. These filling techniques are capable to fill air holes with diameters of a few micrometers. However, the authors strongly believe that the liquid filling into the core size of 280 µm of this proposed PCF can be carried out using one of these techniques.

The considered both circular and hexagonal core of the proposed sensor can be filled up with adulterated fuel samples of different RI, and the sensing characteristics are analyzed by transmitting electromagnetic signal of THz band through the considerate core. Predominantly it is noticed that the RI of pure petrol changes with the percentage of kerosene mix in it. The RI of adulterated petrol with different concentrations of kerosene is shown in Table 1. 

The RS of the sensor depends on how strongly the electromagnetic radiation of THz band interacts with the fuel sample. The absorption coefficient at a given frequency is dependent on that principle and can be determined in accordance with the Beer–Lambert law at an exact frequency as [34];
(1)$$I\left(f\right)=I_{0}\left(f\right)\exp\left[-r\alpha_{mat}l_{c}\right]$$
where, *I(f)* signifies free space propagation intensity of electromagnetic radiation through the core filled with fuel sample, *I_0_(f)* signifies the electromagnetic radiation without any existence of fuel sample, *r* is the relative sensitivity, *f* is the frequency, *l_c_* is the core length, and *α_mat_* is the bulk material absorption coefficient which defines the degree at which the intensity of light reduces as it flows through the material.

The absorbance of the fuel sample to be sensed can be calculated by [34];
(2)$$A=\log\left(\frac{I\left(f\right)}{I_{0}\left(f\right)}\right)=-r\alpha_{mat}l_{c}$$

The sensitivity of any sensor can be defined as the ability to detect any presence of extrinsic material in sample under test. In this case, the ability to detect the variation in RI due to the presence of kerosene concentration in petrol can be referred to as the RS which can be calculated as [34];
(3)$$r=\frac{n_{r}}{n_{eff}}\times P_{c}$$
where *n_r_* denotes the real RI of the fuel sample, *n_eff_* is the real effective RI of the fundamental mode and the rest *P_c_* characterizes the proportion of light interaction with the fuel sample which can be determined by employing Poynting’s theorem as [34];
(4)$$P_{c}=\frac{\int_{Sample\;}R_{e}(E_{x}H_{y}-E_{y}H_{x})dxdy}{\int_{All\;}R_{e}(E_{x}H_{y}-E_{y}H_{x})dxdy}\times 100$$
where *H_x_*, *H_y_* and *E_x_*, *E_y_* are the magnetic and electric field components of the fundamental mode, respectively.

## 3. Results and Discussion

From Equation (3), it can be seen that the RS is inversely and linearly proportional to the effective refractive index and core power fraction of the guided mode, respectively. That’s why the variation of core power fraction and effective mode index for different fuel samples of petrol (concentrations of kerosene is varied from 0% to 100%) over considered THz frequency band. The variation of core power fraction with the frequency with a core diameter of D_C_= 280 µm for both circular and hexagonal cores is shown in Figure 3.

In Figure 3, the solid and dash lines indicate the performance of the proposed sensor for circular and hexagonal core structures, respectively. From Figure 3, it can be perceived that the core power fraction gradually increases with the expansion of operating frequency up to 2 THz. However, for circular core configuration, the core power fraction gradually decreases after 2 THz. However, for the hexagonal core, the fraction of core power shows some steady values after 2 THz. The increment in core power fraction is negligible until 2.9 THz and it started to decrease with increasing further operating frequency.

Figure 4 illustrates the effective mode index variation with the frequency. From Figure 4, it can be seen that the real part of the effective mode index increases exponentially as the frequency increases for both hexagonal and circular core structures. Since there is an inverse relationship between RS and effective mode index, the RS of the sensor would decrease with the expansion of the frequency range. After considering the variation of core power fraction and effective mode index with the frequency, we choose f = 2 THz as the optimum operating frequency for the proposed sensor.

In order to ensure the highest possible performance, the RS is investigated with the variation of designed parameters at different frequencies. First of all, the RS of the sensor is examined with the variation of core diameter. In this analysis, the considered circular and hexagonal core diameter of the sensor is changed from 260 to 290 µm as shown in Figure 5. Since RS is directly proportional to the RI of the fuel sample, the proposed sensor exhibits higher RS for 100% concentration of kerosene than others. In addition, the RS of the proposed sensor with hexagonal core formation at any particular core diameter is higher than circular core formation. On top of that, the RS increases with the expansion of core diameter as a larger core diameter facilitate better electromagnetic radiation interaction with fuel samples. However, a larger core diameter decreases the core strut thickness (z, the gap between the core and cladding) which leads to the proposed sensor being frailer. Hence, to trade-off between the sensor strength and performance, the D_c_ is chosen of 280 µm which offers the z of 10 µm. 

Figure 6 depicts the effects on relative sensitivity over the variation of cladding air hole diameter. This analysis is carried out with a fixed core diameter of 280 µm. From Figure 6, it can be observed that RS is almost constant over the varied air hole diameter of 628.43 to 643.43 µm. Besides that, for any specific air hole diameter, the sensitivity response of the proposed sensor with hexagonal core formation is higher than circular core formation. Since larger air hole diameter increases the overall diameter of the proposed sensor and variation of sensitivity is negligible over varied ranges, we set D = 643.34 µm as the optimum air hole diameter.

The consequences of varying core strut thickness (the adjacent gap between two consecutive air holes, t) on relative sensitivity is shown in Figure 7. During this analysis, we have kept the core and air hole diameter fixed at 280 and 643.43 µm, respectively. The result shows that the sensitivity decreases slightly for all analyzed fuel samples with the growth of core strut thickness. However smaller core strut thickness results in degradation in sensor performance in case of rigidity. Therefore, considering fabrication feasibility and structural rigidity we set, t = 15 µm as the optimum core strut thickness. However, the RS of the proposed sensor with the variation of D, D_c_, and t for 100% Kerosene is summarized at Table 2.

However, we choose the core diameter, D_c_ = 280 µm, air hole diameter, D = 643.34 µm, core strut thickness, t = 15 µm, and the gap between adjacent cladding air hole, z = 10 µm as the optimum design parameters. The electric field distribution of the designed sensor at optimum design consideration is presented in Figure 8. Figure 8a,b show the fundamental mode distribution of neat samples of petrol in circular and hexagonal core formation, respectively. From these mode distributions, it can be seen that at optimum design conditions modes are tightly bound in the core section area and more sensitive to any changes in RI of fuel samples. Due to this well-confined radiation in the core section, the core is the perfect spot to fill with adulterated fuel samples to sense any variation in the RI of the sample under test.

Finally, the RS variation of the suggested PCF sensor with frequency is shown in Figure 9. If we consider the core configuration individually it can be observed that for a circular core, the sensitivity responses for fuel samples of different RI increase quickly from 1.0 to 1.5 THz, then reaches its peak value around 1.7 THz, and after that, it swiftly decreases with the increasing frequency. For hexagonal core consideration, the RS increases gradually from 1.0 to 1.8 THz, reaches its peak value around 2.0 THz, remains constant from 2.0 to 2.3 THz, and after that, it decreases slightly with the expansion of frequency. The RS response of the reported sensor at optimum design consideration is shown in Table 3.

Now the associated losses that occur at the time transmitting electromagnetic radiation through the proposed oil adulteration sensor is characterized. There are mainly two types of losses that occur during the propagation of electromagnetic radiation and these are CL and EML.

CL exits in optical waveguides because some part of guided radiation penetrates the cladding region causing leakage and it determines the confinement capability of a waveguide. The CL can be calculated as [34];
(5)$$\mathrm{CL}=8.686\times k_{0}\times Im\left(n_{eff}\right)$$
where *k_0_* indicates the free space vector and *Im(n_eff_)* denotes the imaginary effective refractive index of fuel sample. 

Figure 10a,b illustrates the CL profile of the proposed sensor at optimum design considerations for hexagonal and circular cores, respectively. The outcomes show that at first CL decreases with the increment in frequency until 1.6 THz both for hexagonal and circular core configurations. Although after that there happen to be some fluctuations in the CL profile as the frequency increases. However, the CL is obtained under dB/m in the entire operating frequency, and at the optimum frequency (2 THz) the CL is less than 10^−10^ dB/m for both circular and hexagonal core formation for the variation of kerosene from 0% to 100%.

The effective material loss (EML) refers to how much light energy the core material itself absorbs and it is the most sovereign type of propagation loss that affects the performance of the optical waveguides. It mainly relies on the used fiber material and can be calculated as [34];
(6)$$\mathrm{EML}=\sqrt{\frac{\varepsilon_{0}}{\mu_{0}}}\;\frac{\int_{A_{mat}}^{\;}\eta \alpha_{mat}{\left|E\right|}^{2}dA}{2\int_{A_{ll}}^{\;}S_{z}dA}$$
where *ε*_0_ is the permittivity and *μ*_0_ is the permeability of vacuum, *η* is the RI of polymer, *α_mat_* is the bulk material absorption loss and *S_z_* is the *z*-component of the poynting vector and *S_z_* = *Re*(*E* × *H**)*z* ⋅ Here *H* and *E* are magnetic and electric field components, respectively.

Figure 11 depicts the variation of EML with the frequency at different adulteration levels of the suggested sensor. For the reported sensor with the circular core, the result shows that EML decreases slowly as the operating frequency increases after that, the EML is approximately constant at the range of 1.6 to 2 THz and then it gradually increases with increasing frequency. For hexagonal core consideration, EML tends to decrease until 1.9 THz after that, the EML is nearly constant at the range of 2.0 to 2.5 THz and then it slowly increases with the growth of operating frequency. At 2 THz, the reported sensor exhibits low EML of 0.030–0.026 dB/cm and 0.063–0.053 dB/cm for hexagonal and circular core configurations, respectively, for low to a high percentage of kerosene concentration (% *v*/*v*) in Petrol.

Now the total loss of the suggested oil adulteration sensor is calculated by adding CL and EML and is shown in Figure 12. From Figure 12, it can be observed that total loss follows the track of EML as the CL is very low compared to EML. At 2 THz, the reported sensor exhibits very low total loss which is less than 0.07 dB/cm for both hexagonal and circular core configurations, respectively for different adulteration levels.

Moreover, another two important parameters related to the sensor performance are numerical aperture (NA) and marcuse spot size (MSS). The NA of the reported sensor can be quantified as [42];
(7)$$\mathrm{NA}=\frac{1}{\sqrt{1+\frac{\pi A_{eff}f^{2}}{c^{2}}}}$$
where, *f* is the frequency, *c* is the velocity of light and *A_eff_* is the effective area. 

The NA of the reported sensor for both hexagonal and circular cores is displayed in Figure 13 for considered ranges of THz frequency. From Figure 13, it is clear that the NA shows a downward trend in the entire operating frequency. Note that for broad sensing applications, a wide NA is always desirable. However, at 2 THz, the NA are 0.31 and 0.32 for hexagonal and circular core configurations, respectively.

Finally, the marcuse spot size (MSS) which is a measure of mode field radius of the proposed oil adulteration sensor, can be evaluated by the following expression [42];
(8)$$W_{eff}=r\left.\times \left(0.65+\frac{1.619}{V^{1.5}}+\frac{2.879}{V^{6}}\right.\right)$$
where, *r* and *V* denote hollow-core radius and normalized frequency, respectively.

For sensing applications, a big spot size is always preferable to maximize the interaction between the radiation signal and the fuel samples, consequently maximizing sensitivity. From Figure 14 it can be observed that, for hexagonal core, spot size decreases swiftly as the operating frequency increases until 2 THz, then the spot size tends to be decreasing slowly following the rest of the frequency band. For circular core, spot size decreases with the growth of operating frequency. At optimum design conditions, the average spot size is 194 and 191 µm for hexagonal and circular core configurations, respectively, for all concentrations of kerosene in petrol.

The reported sensor performance is compared with the prior sensor which is shown in Table 4. Considering sensing performance, transmission loss profiles at optimum design conditions as well as fabrication feasibility this proposed LI-PCF petrol oil adulteration sensor, is well in advance of others.

## 4. Conclusions

In this work, a grapefruit-shaped PCF with two different core configurations (circular and hexagonal) is proposed for the detection of adulterated fuel, and the sensor performance is carried out in the THz frequency that ranges from 1 to 3 THz. However, in optimum frequency (2 THz), the sensor with hexagonal and circular core configurations achieves high sensitivity response of 97.27% and 93.50%, respectively, CL under 10^−10^ dB/m, and EML and total loss both are less than 0.07 dB/cm. The proposed sensor exhibits a large effective modal area, high numerical aperture, and large spot size at optimum design conditions. Moreover, this sensor can be implemented in practice by existing fabrication methodologies and consequently making this petrol adulteration sensor a promising candidate in commercial and industrial sensing.

## Figures and Tables

**Figure 1 polymers-15-00151-f001:**
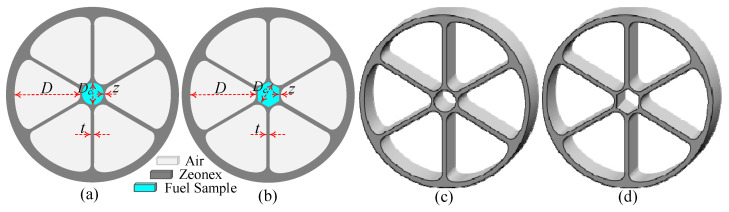
2D view of (**a**) circular, and (**b**) hexagonal core configuration, and 3D view of (**c**) circular, and (**d**) hexagonal core configuration and meshing of (**e**) circular, and (**f**) hexagonal core configuration of the proposed grapefruit hollow-core PCF.

**Figure 2 polymers-15-00151-f002:**
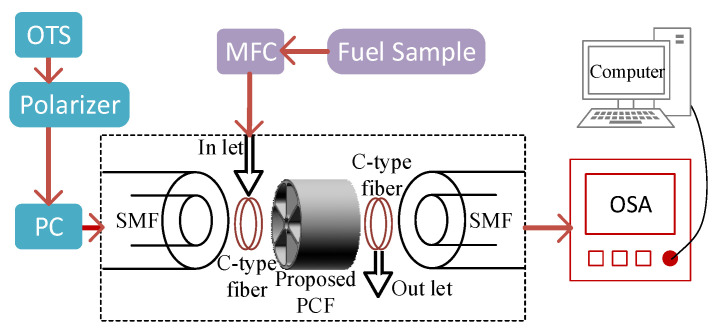
An operational approach for the pragmatic execution of the proposed sensor.

**Figure 3 polymers-15-00151-f003:**
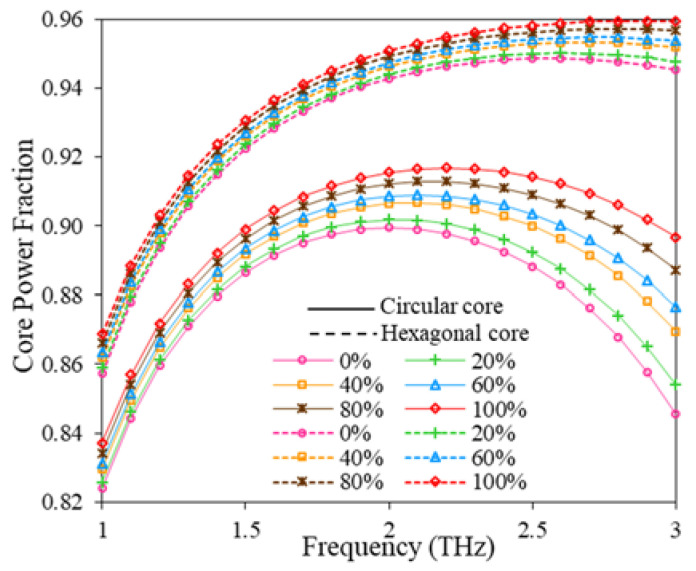
Core power fraction for hexagonal and circular core with the change of frequency.

**Figure 4 polymers-15-00151-f004:**
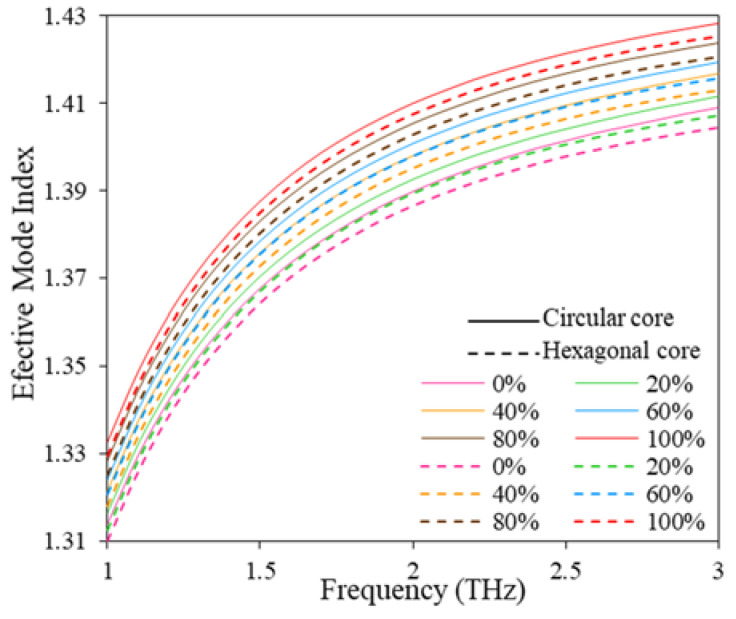
Effective mode index for hexagonal and circular core with the change of frequency.

**Figure 5 polymers-15-00151-f005:**
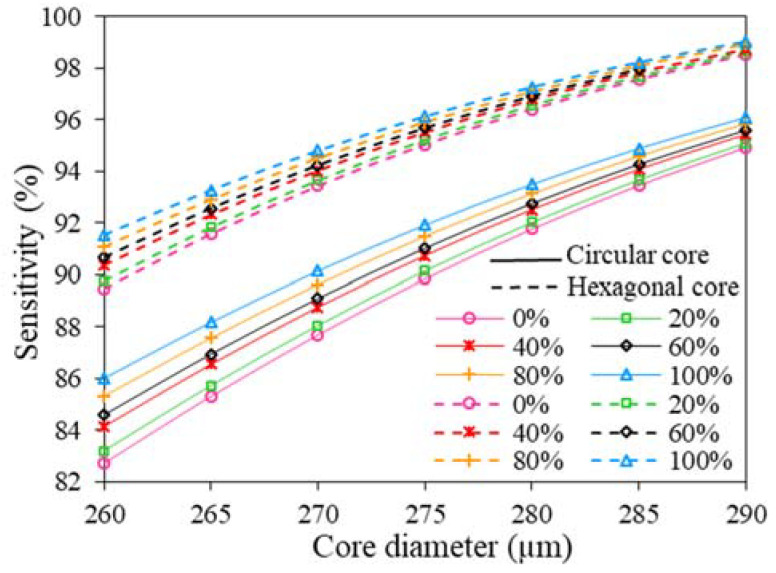
The variation of RS with Dc of hexagonal and circular core at 2 THz.

**Figure 6 polymers-15-00151-f006:**
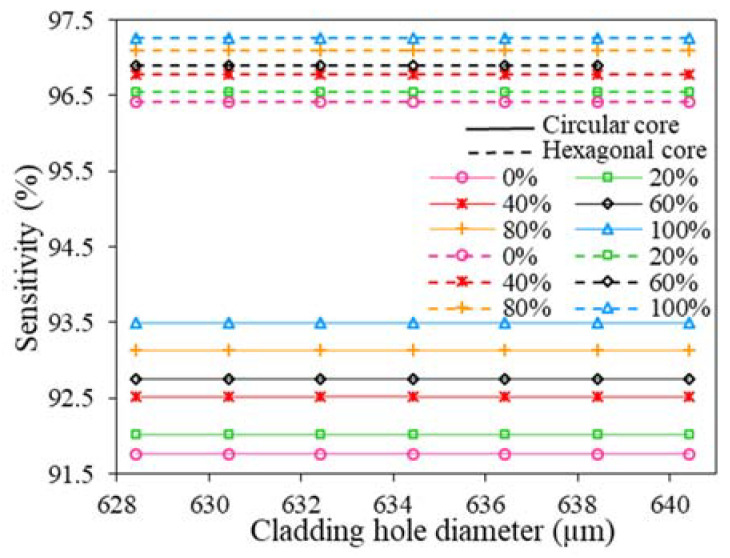
The variation of RS with D of hexagonal and circular core at 2 THz.

**Figure 7 polymers-15-00151-f007:**
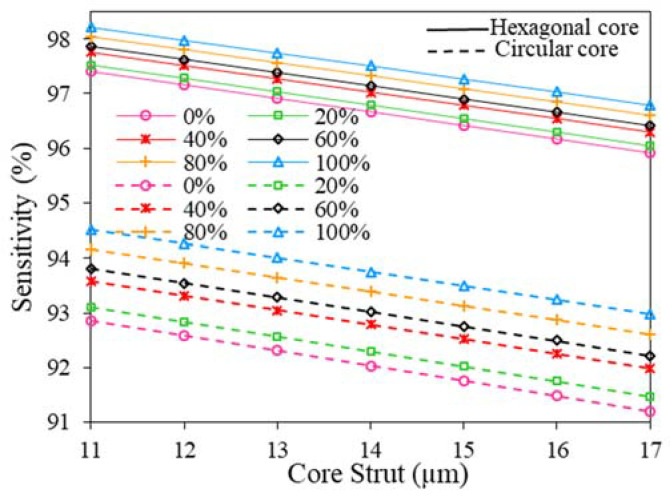
The variation of RS with *t* of hexagonal and circular core at 2 THz.

**Figure 8 polymers-15-00151-f008:**
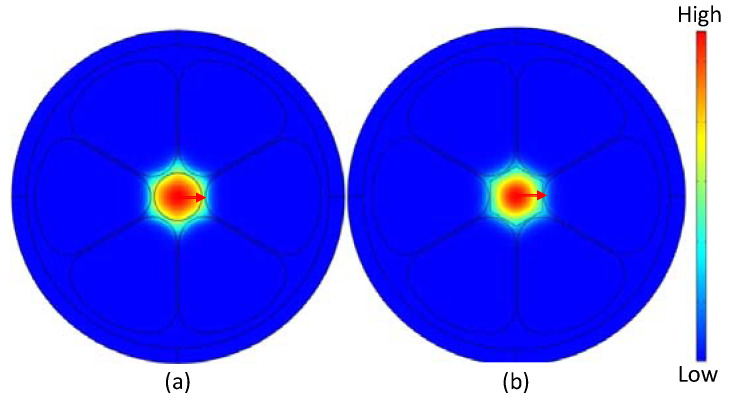
Field pattern of the x-polarization core mode for (**a**) circular, and (**b**) hexagonal core at optimum design conditions.

**Figure 9 polymers-15-00151-f009:**
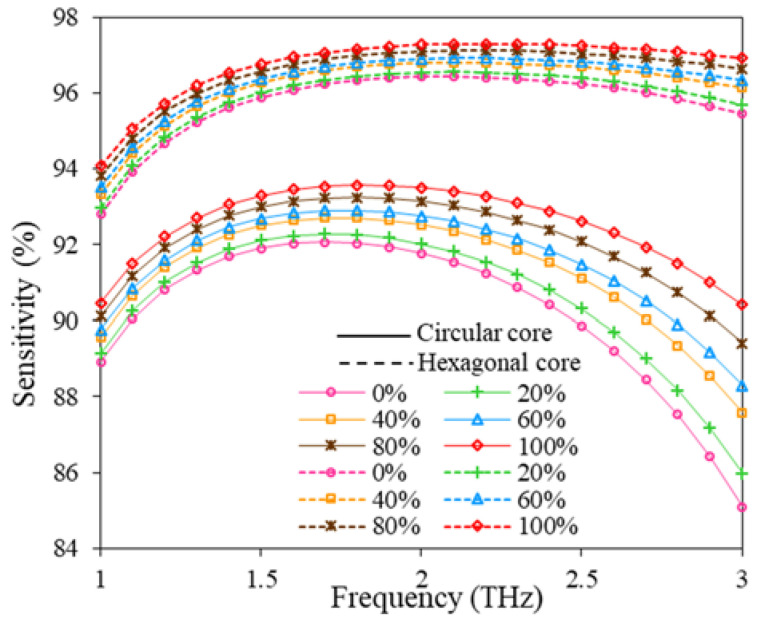
RS variation with the change of frequency for D = 643.34 µm, D_c_ = 280 µm, t = 15 µm, z = 10 µm.

**Figure 10 polymers-15-00151-f010:**
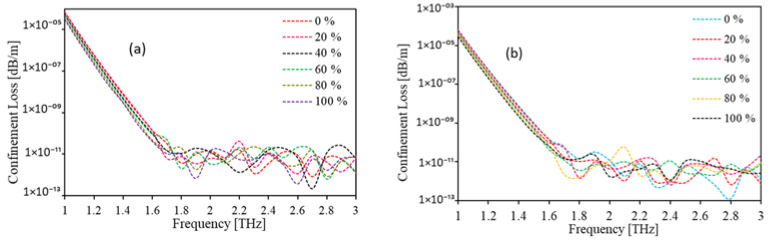
The CL for (**a**) hexagonal, (**b**) circular core with the change of frequency for D = 643.34 µm, D_c_ = 280 µm, t = 15 µm, z = 10 µm.

**Figure 11 polymers-15-00151-f011:**
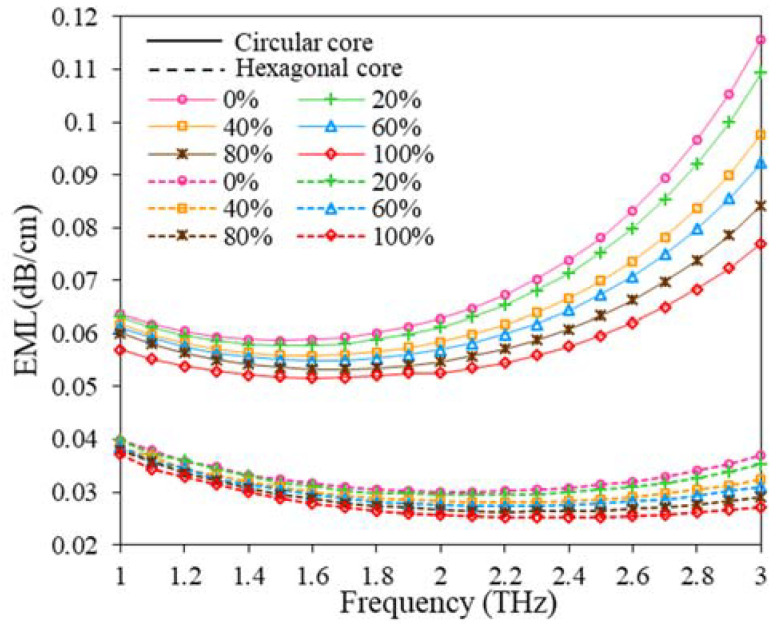
EML variation with the change of frequency for D = 643.34 µm, D_c_ = 280 µm, t = 15 µm, z = 10 µm.

**Figure 12 polymers-15-00151-f012:**
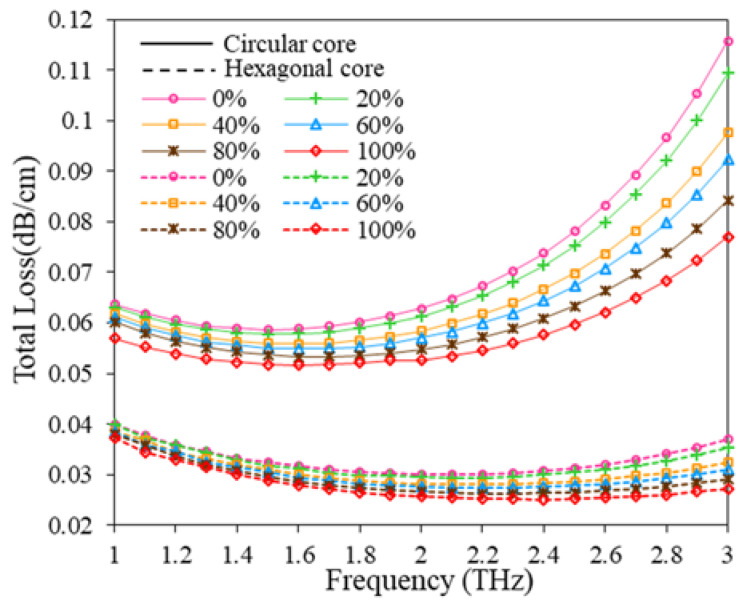
Total loss variation with the change of frequency for D = 643.34 µm, D_c_ = 280 µm, t = 15 µm, z = 10 µm.

**Figure 13 polymers-15-00151-f013:**
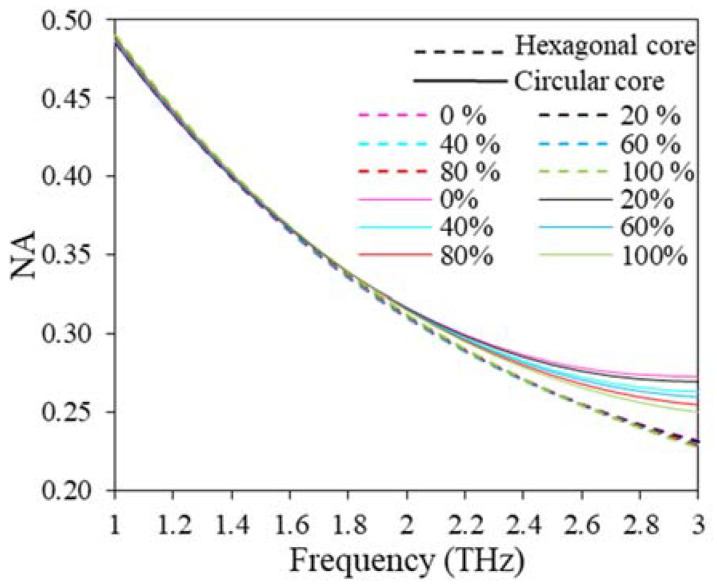
NA variation with the change of frequency for D = 643.34 µm, D_c_ = 280 µm, t = 15 µm, z = 10 µm.

**Figure 14 polymers-15-00151-f014:**
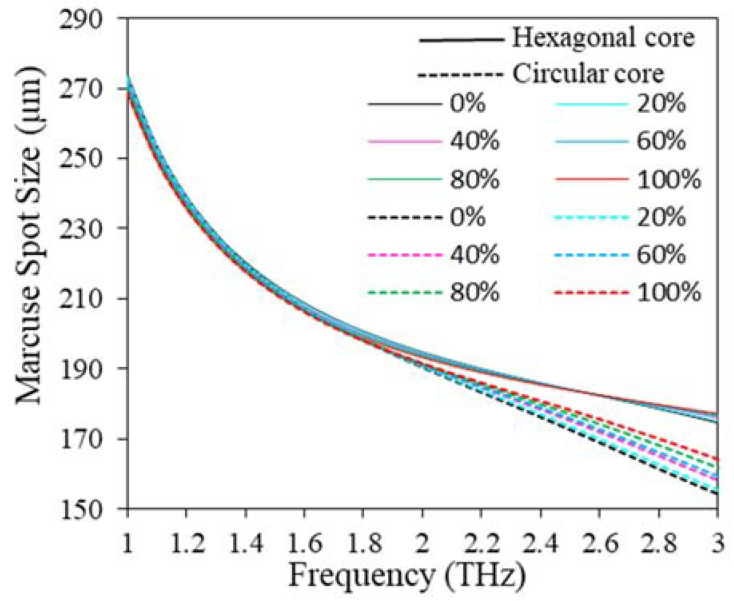
MSS variation with the change of frequency for D = 643.34 µm, D_c_ = 280 µm, t = 15 µm, z = 10 µm.

**Table 1 polymers-15-00151-t001:** The RI of adulterated petrol with different concentrations of kerosene [42].

Percentage of Kerosene (% *v*/*v*) in Petrol	Refractive Index (RI)
0 (Neat sample of Petrol)	1.418
20	1.421
40	1.427
60	1.430
80	1.435
100 (Neat sample of Kerosene)	1.440

**Table 2 polymers-15-00151-t002:** RS of the proposed sensor with the variation of D, D_c_, and t for 100% Kerosene.

Dc (µm)	Sensitivity (%)		D (µm)	Sensitivity (%)		t (µm)	Sensitivity (%)	
	Circular Core	Hexagonal Core		Circular Core	Hexagonal Core		Circular Core	Hexagonal Core
260	86.01527	91.53090548	628.43	93.49685	97.26752	11	94.51338	97.41435
265	88.17886	93.27275048	630.43	93.49685	97.26752	12	94.26435	97.37572
270	90.15067	94.8031505	632.43	93.49685	97.26752	13	94.00766	97.34298
275	91.92119	96.1305311	634.43	93.49685	97.26752	14	93.75659	97.30921
280	93.49685	97.26751918	636.43	93.49685	97.26752	15	93.49685	97.26752
285	94.88405	98.22992395	638.43	93.49685	97.26752	16	93.24272	97.23274
290	96.08930	99.03263884	640.43	93.49685	97.26752	17	92.98095	97.18900

**Table 3 polymers-15-00151-t003:** RS of the proposed sensor at optimum design conditions.

Kerosene Percentage (% *v*/*v*) in Petrol	RS (%) for Circular Core	RS (%) for Hexagonal Core
0	91.76	96.42
20	92.02	96.54
40	92.52	96.79
60	92.76	96.90
80	93.13	97.09
100	93.50	97.27

**Table 4 polymers-15-00151-t004:** Result comparison with other prior fuel adulteration sensors.

Ref.	Operating Point	Considered Percentage of Kerosene (% *v*/*v*) in Petrol	RS (%)	EML (dB/cm)	CL (dB/m)
[41]	-	10–50	98.36	-	-
[42]	2.8 THz	0–100	89.40	0.095	10^−8^
[57]	1 THz	0–100	96.40	0.014	3.86 × 10^−5^
[58]	2.5 THz	0–100	98.68	0.034	4.12 × 10^−14^
This Work	2.0 THz	0–100	97.27	0.026	10^−10^

## Data Availability

Data underlying the results presented in this paper are not publicly available at this time but may be obtained from the authors upon reasonable request.

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
