# Peer review of "Designing of Hollow Core Grapefruit Fiber Using Cyclo Olefin Polymer for the Detection of Fuel Adulteration in Terahertz Region"

_polymers, 2022, doi:10.3390/polym15010151_

Round 1

Reviewer 1 Report

The authors have introduced a new configuration of optical fiber in their paper titled "Designing of Hollow Core Grapefruit Fiber Using Cyclo Olefin Polymer for the Detection of Fuel Adulteration in Terahertz Region". Many researchers in the related field are carrying out the same kind of analysis where they are proposing new configurations for different detection applications. With the introduced configurations the authors have increased the detection performance to more than 95% accuracy, however other researchers have also reported accuracies of 90-92% which resultantly undermines the importance of this work.This reviewer feels that the paper is of average importance and must consider revision according to the following comments:

1. The authors have told the number of meshes used in the simulations which come out of nowhere. A propoer mesh sensitivity analysis needs to be added to give credence to the simulations.

2. Error analysis need to be done wither with so analytical solution, other numerical analysis of with some reported experimental work.

3. There is no info on tuning parameters for the results, assumptions and input variables used in the simulations It is good to add these details for the better understanding of the reviewers.

4. A small section should be added to indicate the manufacturing challenges of the designed fibres.  

Reviewer 2 Report

In this work,a photonic crystal fiber sensor is proposed to identify the percentage of kerosene in adulterated petrol. The manuscript is clear and well-organized in the article structure. However, some related papers have been published, such as, [1] Ferdous A ,  Anower M S ,  Habib M A . A hybrid structured PCF for fuel adulteration detection in terahertz regime[J]. Sensing and Bio-Sensing Research, 2021, 33(1):100438. [2] Bulbul A M ,  Rashed A ,  El-Hageen H M , et al. Design and numerical analysis of an extremely sensitive PCF-based sensor for detecting kerosene adulteration in petrol and diesel[J]. AEJ - Alexandria Engineering Journal, 2021, 60(6):5419-5430. The author needs further clarification about the advance they made in this field.

1. Table 1 shows the RI of adulterated petrol with different concentrations of kerosene. However, the cited references[44] is not relevant to the RI of adulterated petrol.

2. The electric field direction should be shown in Figure 8.

3. In table 3, there are few parameters and literatures for comparative study.

4. The author mentioned several structural parameters(D, t and Dc) in the second part, but no specific value is given. Are their values variable? Figure 9-14 is drawn based on what parameter(D, t and Dc)? Why these values are selected?

5. The diameter of single mode fiber is about 125um,whether there will be large loss when coupling with the proposed PCF with the diameter of 1.66×10-03 m

6. There are still some details in the manuscript that need to be modified,for example, there are two punctuations at the end of abstract.

Round 2

Reviewer 1 Report

The authors have sufficiently improved the manuscript. There are few typos hence it is strongly recommended to have a thorough check again for English related mistakes.